

Uncovering a Key Predictors for Enhancing Daily Streamflow Simulation Using
Machine Learning
Arash Aghakhani[a,b,*] , David E. Robertson[a], Valentijn R.N. Pauwels[b]
[a] Commonwealth Scientific and Industrial Research Organisation (CSIRO), Clayton 3168, Australia
[b] Department of Civil Engineering, Monash University, Clayton, VIC 3800, Australia
Correspondence: Arash Aghakhani, arash.aghakhani86@gmail.com
Abstract
The sequence of droughts and wetter periods in Australia poses challenges for long-term hydrologic
modelling. This paper develops a novel machine learning-based approach to uncover key predictors that
improve daily streamflow predictions during and after the Millennium drought (1997 to 2009) in 39
gauged sub-catchments in Western Victoria, Australia.
For this purpose, a hybrid approach is adopted, combining simulations from the GR4J hydrological
model with physical data as forcing (predictors) for multiple ML algorithms to identify the key
predictors for improving streamflow prediction. GR4J is a widely used operational hydrological model
in Australia. ML models including predictors representing long-term runoff coefficient and short-term
runoff and rainfall showed the greatest improvement in streamflow predictions, particularly for low
flows. This suggests that GR4J has limited ability to capture short/long-term persistence and therefore
model enhancement should focus on these shortcomings. All ML algorithms resulted in improved
streamflow prediction, with Multilayer Perceptron (MLP) consistently yielding the highest Nash
Sutcliffe Efficiency, and Random Forest showing the strongest improvement in terms of low-flow
prediction. Long-term runoff coefficient and machine learning were most effective in catchments with
lower long-term runoff coefficients. Overall, this study provides insights for water resources
management in drought-prone regions, highlighting the key predictors in the combination of ML and
hydrological modelling to improve streamflow predictions during and after droughts.





Keywords: Machine learning, streamflow prediction, Millennium drought, Climate change, Hydrological model, long/short-term runoff coefficient

## 1  Introduction

Climate change is projected to increase the frequency and severity of droughts in many parts of the world due to reductions in rainfall (Fowler et al., 2022). Analysis of historical multi-year droughts can therefore provide insights and understanding of hydrological responses likely to be observed under future climate changes (Fowler et al., 2022). Australia's Millennium drought (1997 to 2009) serves as a noteworthy example(Van Dijk et al., 2013). During the Millenium drought, a substantial portion of the southern part of Australia faced an extended period of dry conditions. The impact was particularly severe in densely populated areas of Southeastern and Southwestern Australia (Bureau of Meteorology, 2024) (Wallis et al., 2011)(Southwest Victoria alliance-Advocacy Priorities 2021/22, 2024)(Primary Production Landscapes of Victoria - Northwest Victoria, (2024)).During the Millennium drought, soil drying and declining water tables harmed flora and fauna, leading to more wildfires and dust storms (Heberger, 2012). The sheep population during the Millennium drought in Australia declined by half (Heberger, 2012). Water shortages due to the Millennium drought led to decreased water allocations for irrigation, imposed water restrictions in urban areas, and prompted considerable investments in infrastructure, such as the construction of desalination plants and pipeline installation (Van Dijk et al., 2013).

Southwestern Victoria, as part of southeastern Australia severely impacted during the Millennium Drought, reflects the nation's broader struggle with prolonged dry conditions. During the Millenium drought, alterations, or shifts (Saft et al., 2015), in rainfall–runoff relationships were observed in numerous Victorian catchments but were absent in others (Fowler et al., 2022). These shifts in rainfall-runoff relationships are a form of hydrologic non-stationary. Non-stationarity in hydrology refers to the phenomenon where the statistical properties of hydrological processes, such as rainfall patterns and runoff relationships, change over time. In other words, the influence of forcing variables in one period will differ from those experienced in a separate period (Slater et al., 2021). During the Millennium



drought reductions in streamflow were larger than initially anticipated based on the observed rainfall
reductions (van Rensch et al., 2023). The physical mechanisms behind these shifts are not well
understood (Fowler et al., 2022). According to Peterson et al. (2021), the transitions towards reduced
streamflow can exhibit remarkable persistence, as certain watersheds appear to maintain a changed state
even after returning to nearly average climate conditions. In these persisting cases, a year of typical
rainfall now results in lower streamflow compared to pre-drought levels.
Streamflow prediction is challenging due to its high complexity, non-stationarity, and non-linearity of
hydrological processes (Yaseen et al., 2015). Many Process-Based (PB) hydrological models have been
developed for streamflow prediction. However, due to the uncertainties in the formulation and
parameterisation of the physical processes, and also the errors in the forcings and validation data, these
models exhibit limited capability in capturing the non-stationary in hydrological datasets (Cheng et al.,
2020). Conceptual hydrological models are widely used for climate change impact analyses due to their
higher computational efficiency, facilitating rapid assessments across multiple catchments and for long-
term simulations (Zheng et al., 2024) However, their simplified representation of the physical processes
potentially overlooks the ways within-catchment hydrological processes could intensify or alleviate
changes in runoff generation under a changing climate (Robertson et al., 2023). Since the physical
mechanisms behind the shifts in rainfall-runoff relationships are not well understood, such shifts in
response to extended drought are challenging to predict. Furthermore, it is widely accepted that the use
of hydrological models for prediction in changing conditions remains a significant concern in hydrology
(Blöschl et al., 2019).
Model validation against a diverse range of conditions is a crucial step in building robust models
(Fowler et al., 2022). Hydrological models developed and calibrated based on pre-drought data are not
capable of properly predicting the streamflow during the drought (Chiew et al., 2014). Fowler et al.
(2021) studied the performance of the GR4J hydrological model in Victoria's catchments for the
Millennium drought and suggested that limitations in conceptual hydrological models capabilities to
project runoff under a drying climate might be due to implicit upper limits on the soil moisture deficit.
Chiew et al. (2014) modelled 20 catchments in South-Eastern Australia, using the SIMHYD model, and



found that pre-drought optimised parameter values resulted in very poor Millennium drought daily
streamflow simulations. They stated that this poor performance is related to the hydrologic non-
stationarity.
Machine learning models have proven highly effective in modelling natural systems because they do
not make prior assumptions on the form of equations describing physical processes that are necessary
in traditional models (Reis et al., 2021). Over the past two decades, there has been substantial
advancement in the application of artificial intelligence and machine learning (ML) methods to predict
non-linear hydrological responses (Pradhan et al., 2020), and generate valuable insights into the
physical mechanisms behind hydrologic non-stationarities (Hao and Bai, 2023). The popularity of ML
models for streamflow prediction is related to their relative ease of application, competitive
computational performance and prediction accuracy (Pham et al., 2021). Machine learning methods are
also less rigid in assumptions used to characterize the probability distribution of model errors than other
methods, including statistical modelling. PB and ML approaches can complement each other with
respect to their strengths and limitations. Hydrologic processes that preserve the mass and energy
balance can be captured by a PB model, while other processes not incorporated in a PB model can be
covered by the data-driven ML algorithm. Thus, an effective strategy is a hybrid combination of both
approaches to improve hydrologic simulation (Konapala et al., 2020).
In ML, computers learn from data patterns to make predictions without the need for code development
by the user. Input data (predictors) are used to predict output variables. Various machine learning
methodologies have been employed in hydrological research for predicting streamflow, (Deo and Şahin,
2016; Petty and Dhingra, 2018; Yaseen et al., 2016). These approaches are based on diverse techniques
such as Linear Regression, Multiple Linear Regression (MLR), Ridge regression, Gradient Boosting
and its variants, neural networks (NN), specifically Multilayer Perceptron (MLP), and long short-term
memory (LSTM), support vector machines (SVM), Decision tree, and Random Forest (RF), among
others. Recent research indicates that ML models exhibit robustness and efficiency, often surpassing
the performance of traditional hydrological models, including both conceptual and physically-based
approaches, (Akusok et al., 2015; Rozos et al., 2022; Worland et al., 2018).



Numerous reviews of the use of artificial intelligence and machine learning for hydrological
applications have been published (Ng et al., 2023; Mohammadi, 2021; Yaseen et al., 2015). These
reviews have all concluded that ML methods are highly suited to runoff prediction and forecasting
applications due to their ability to represent non-linear and highly complex relationships between
prediction variables and their predictors, and integrate a large range of complex and noisy datasets
published (Ng et al., 2023; Mohammadi, 2021; Yaseen et al., 2015)
However, no standardised guidance exists for selecting ML methods to achieve optimal simulation
accuracy (Hao and Bai, 2023) and choosing a single ML algorithm with high predictive accuracy is
challenging, because algorithm performance depends on the specific datasets used and problem being
addressed (Hao and Bai, 2023; Worland et al., 2018). A number of studies have reported varying
outcomes regarding the comparison of the performance of different algorithms (Section 2.2.2.8). The
algorithm is not the only factor of influence on the prediction results, the quality and quantity of the
input data (predictors) has been shown to have a strong impact as well.
Moosavi et al. (2022) highlighted that the volume of input data (predictors) had stronger impact on the
prediction accuracy in comparison to the model type. The lack of adequate predictor selection, or the
use of an excessive number of predictors, affects the precision and stability of prediction models. This
situation also necessitates a substantial amount of memory and computational resources for training and
validation procedures (Reis et al., 2021). Therefore, identifying a suitable set of predictors is a crucial
step in the ML modelling process. ML algorithms may produce less accurate and less interpretable
results when working with insufficient data, or with data containing irrelevant or redundant information.
In hydrologic problems, nonlinear approaches are often preferred for determining the type of model
inputs, given the demonstrated highly non-linear nature of the input–output relationships (Lima et al.,

127    2016).

Published studies have primarily focused on enhancing streamflow forecasting at the sub-catchment
scale through machine learning, rather than finding the predictors that lead to the greatest improvement
in the performance of predictions across a range of catchments. Additionally, there is a literature gap on
how machine learning can be used to improve hydrological models and to identify potential



enhancements that could be globally applicable. Therefore, in this study, ML is employed using a
combination of hydrological model outputs and physical data as predictors to investigate possible
improvements in prediction performance and identify components missing in the hydrological model
conceptualisation.  For this purpose, GR4J, a simple yet efficient hydrological model (Arsenault et al.,
2019), appropriate for a range of climate conditions and commonly used in Australia (Stephens et al.,
2019) is used in this study.
The objectives of this study are: (1) to explore the effectiveness of using a combination of GR4J and
physical data as predictors for machine learning algorithms for streamflow prediction in Western
Victoria, Australia, during the drought and the post-drought periods, based on pre-drought calibration,
and (2) to reveal the catchment data that are overlooked or are too uncertain for the employed
hydrological model structure or configuration. This can be achieved by identifying the predictors with
a significant impact on the improvement of the model results, incorporating them as essential physical
parameters in the conceptual hydrological model, or by making alterations to the structure of the
hydrological model.
The paper is structured as follows: Section 2 describes the case study area and the methodology used in
this study. Section 3 presents results and discussion of the performance of hydrological model and
proposed ML models, followed by section 4 that summarises the key findings and conclusions from the
paper.



## 2 Materials and methods

### 2.1 Case study

For this case study we use streamflow data for 39 gauged sub-catchments of 9 major catchments in western Victoria, Australia. This region depends significantly on agriculture, with approximately 81% of the land use having been developed for agricultural uses (Wallis et al., 2011), and plays a vital role in Australia's agricultural and forestry sectors. It contributes significantly to the nation's dairy production generating 2.3 billion Australian dollars, or 27% of the total national value, annually. In addition, it covers 17% of Australia's forest plantations (Southwest Victoria alliance-Advocacy Priorities 2021/22, 2024). Northwestern Victoria contains 40% of Victoria's dryland agriculture focusing mainly used for cereal cropping. Another significant land use in this area is grazing and production from native vegetation, accounting for 13% of Victoria land area (Primary Production Landscapes of Victoria - Northwest Victoria, 2024). This region has been chosen due to the notable influence of the Millennium drought on the streamflow and its importance on Victoria's economy.

Figure 1 shows an overview of the study area. The study sub-catchment range in size from 4.5 km² to approximately 2,677 km². The majority of the river runoff occurs during the austral winter and spring, primarily attributed to rainfall. The details of the sub-catchments and stream gauge stations can be found in Appendix (Table A. 1 and Figure A. 1). Streamflow data were extracted from the Bureau of Meteorology (BoM) Hydrologic Reference Stations (Hydrologic Reference Stations, 2024). Streamflow data are quality controlled using the quality flags provided, removing observations characterised as estimates, poor or unknown quality (Zhang et al., 2016). Catchment average forcing data, specific are derived by taking area-weighted averages of the Australian Gridded Climate Data (Jones et al., 2009), rainfall and AWRA potential evapotranspiration (Frost, 2018).



## 2.2 Methods


To quantify the impact of the Millennium drought, changes in the mean daily rainfall and mean daily
streamflow were calculated using the following equations:

$$Rainfall\ change = \frac{rainfall_{P2} - rainfall_{P1}}{rainfall_{P1}} \times 100 \qquad \textit{Eq. 1}$$


$$Streamflow\ change = \frac{Streamflow_{P2} - Streamflow_{P1}}{Streamflow_{P1}} \times 100 \qquad \textit{Eq. 2}$$


Where:
Rainfall$_{P1}$: mean daily rainfall in the period from 1988-01-01 to 1996-12-31 (pre-drought)
Rainfall$_{P2}$: mean daily rainfall in the period from 1997-01-01 to 2018-12-31 (drought and post-drought)
Streamflow$_{P1}$: mean daily Streamflow in the period from 1988-01-01 to 1996-12-31 (pre-drought)
Streamflow$_{P2}$: mean daily Streamflow in the period from 1997-01-01 to 2018-12-31 (drought and post-
drought)
Since this study investigates the integration of results from a conceptual hydrological model with ML
algorithms, the remainder of this section focuses on a general discussion on both the conceptual
hydrological model and ML algorithms. Subsequently, the explanation of the proposed models is
provided for comprehensive understanding of the combined approach used in this research.



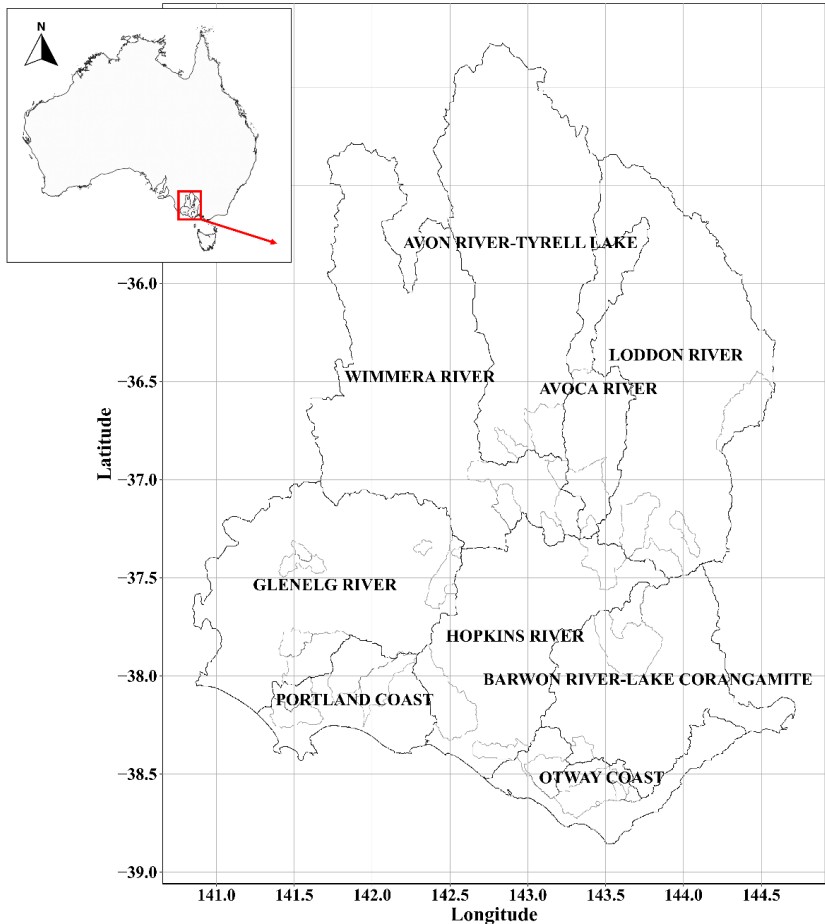


191        *Figure 1. Overview of the catchments and sub-catchments used in this study (projection: EPSG:4326).*

### 2.2.1    Hydrological models
The hydrological model used as a predictor for the ML models in this study is GR4J (Perrin et al., 2003).
A schematic of GR4J is shown in Figure 2. This model is a four-parameter model that has been
demonstrated to perform well in Australia. Model calibration and simulation was undertaken using the
SWIFT2 (Short-term Water Information and Forecasting Tools) software (Perraud et al., 2015). The
GR4J model parameters {X1, X2, X3 and X4} were calibrated by maximising the Nash-Sutcliffe
Efficiency (NSE) using the Shuffled Complex Evolution  (Aryal et al., 2020; Duan et al., 1994; Perrin
et al., 2003).



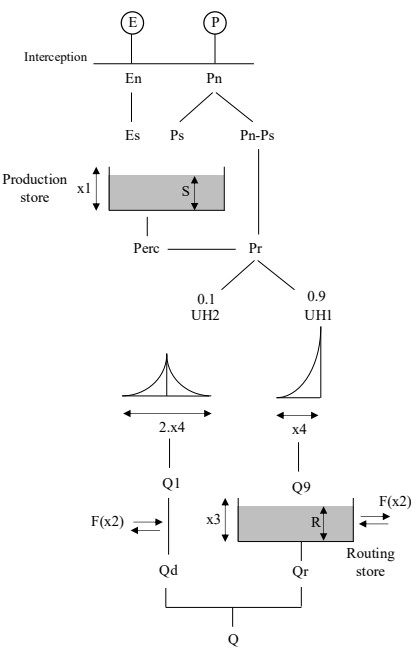


*Figure 2 Diagram of GR4J hydrological model, with parameters derived from* (Perrin et al., 2003)

### 2.2.2   Machine learning algorithms

Various machine learning algorithms are assessed in this study, namely Linear Regression, Ridge Regression, Support Vector Regression (SVR), Decision Tree, Random Forest, Gradient Boosting, and MLP. While all algorithms improved streamflow prediction accuracy, only the results of Random Forest, Gradient Boosting, and MLP are presented as they produced the best performing predictions (Ayzel et al., 2021; Lees et al., 2021; Rozos et al., 2022).

#### 2.2.2.1   *Random Forest regression*

Random Forest regression generates predictions from an ensemble of regression relationships fitted over subsets of the predictor space that are defined using decision trees. This algorithm enhances the stability of decision trees by randomly sampling from the training data with replacement and aggregating the results. The combination of diverse trees also reduces prediction errors. This ensemble method is referred to as Random Forest, given that numerous trees are grown in a 'random' manner (Li et al., 2020). Random Forest is a simple and fast algorithm and one of the most powerful statistical



learning methods. It has the ability to work with very large datasets and is widely used in hydrological
applications (Roy and Larocque, 2012; Szczepanek, 2022). Random Forest avoids assuming a global
functional relationship and uses a local fitting approach through recursive partitioning. It exhibits
flexibility regarding the distributional assumptions of model residuals (Li et al., 2020). This algorithm
decreases variance by averaging decisions across diverse classifiers. Additionally, it diminishes bias
when individual classifiers are adequately complex for the given subset of features (Yeturu, 2020).
Random forest is considered in this study since many previous comparisons stated its remarkable
performance (Bentéjac et al., 2021) and its key advantage in avoiding overfitting issues through an
ensemble of permutations. Additionally, RF model evaluates the importance of predictor variables,
enhancing interpretability compared to methods like artificial neural networks (ANNs) and SVR
(Tongal and Booij, 2018)
*2.2.2.2    Gradient Boosting*
Gradient boosting algorithm is widely recognised as one of the most powerful algorithms (Wang et al.,
2022) which can fit complex nonlinear relationships (He et al., 2020). Gradient boosting iteratively
combines weak learners, which are slightly better than random, to form a suitable learner. This
technique is commonly used for regression tasks. The objective of gradient boosting, when applied to
a training dataset, is to minimise the expected value of a specified loss function to approximate the data
(Bentéjac et al., 2021). Gradient boosting algorithms are well-known for their ability to effectively
handle missing data and intricate relationships (Kumar et al., 2023). This method is commonly
employed with decision trees of a fixed size as the base learners. Boosted decision trees are widely
recognised as among the most effective prediction algorithms available today (Biau et al., 2019). In this
study, gradient boosting is applied as a method for addressing a regression problem.
*2.2.2.3    MLP*
Multilayer Perceptron (MLP) is the most popular type of neural network in hydrology, renowned for its
excellent performance in predicting streamflow (Boucher et al., 2020; Mohammadi et al., 2020;
Rahimzad et al., 2021). This algorithm is a feedforward artificial neural network, which consists of
interconnected nodes organised into three types of layers: input, hidden, and output. The input layer



transmits input values to the hidden layer without performing any operations on the input signal. The
hidden layer processes signals from the input layer nodes, transforming them into signals distributed to
all output nodes. The output nodes, in turn, further transform these signals into final outputs. This
architecture is particularly effective for addressing non-linear problems (Onyari and Ilunga, 2013). In
this study the MLP had 1 hidden layer and 100 neurons.

### 2.2.3 Model application

An overall schematic of the applied methodology is provided in Figure 3. Data were divided into
training and test periods that extended from January 1, 1988, through December 31, 1996, representing
the pre-drought period. Independent predictions were performed for the period January 1, 1997, through
December 31, 2018, covering the Millennium drought and post-drought periods. Calibration and
optimisation were conducted for the training and test periods only, and the subsequent simulation was
carried out for the prediction period.
In the implementation of the machine learning analysis, different combinations of various predictors
were considered in addition to the outcomes of the hydrological model. These predictors include
rainfall, potential evapotranspiration, short-term streamflow memory, short-term rainfall memory, and
long-term runoff coefficient. Short-term streamflow memory is characterised by the average streamflow
over the preceding two days. Similarly, short-term rainfall memory is characterised by the average
rainfall over the previous two days, offering a snapshot of recent precipitation patterns. The short-term
rainfall and streamflow are expected to provide insight into the hydrological responses lag. The long-
term runoff coefficient was derived by evaluating the ratio of cumulative runoff to cumulative rainfall
for a five-year window leading up to each date. The long-term runoff coefficient is assumed to be
associated with changes in catchment characteristics and particularly those related to surface–
groundwater interactions (Robertson et al., 2023).
Eight ML models were assessed through different combinations of predictors, as outlined in Figure 3.
For models 5 and model 6, an intentional effort was undertaken to account for the influence of long-
term runoff coefficient on the estimation of the fraction of the rainfall contributing to streamflow. This



is achieved by multiplying the derived ratio of the five-year streamflow to precipitation, with both the
rainfall and short-term rainfall memory components.
For each model, all ML algorithms listed in section 2.2.2 were trained using the SKlearn package in
python, using data for the period January 1, 1988, through December 31, 1996. The objective function
of all algorithms is squared error. The ML training process requires specification of training and testing
subsets. The division of the data into training and testing periods can have a material impact on the
fitted algorithm and its predictive performance. For this study, we adopted an 80:20 ratio of training to
testing periods. The ML model training process was repeated ten times, each time randomly allocating
data for the period January 1, 1988, through December 31, 1996 to the training and testing periods. All
ten of the fitted ML models were retained to form an ensemble.
The fitted ML models were then used to generate predictions for the period January 1, 1997, through
December 31, 2018. When the ML models are in predictive mode, the predictors representing long-term
and short-term streamflow memories can be derived from observations. Using streamflow observations,
the ML model's predictions are continually adjusted by the most recent observation, and predictions are
equivalent to lead-one forecasts forced by perfect rainfall forecasts. The process begins by initialising
the model with historical data. Then, the model predicts streamflow for the first day using the most
recent observations. After each prediction, the observed streamflow for that day is integrated into the
model as input for the following day prediction. This iterative process continues for the prediction of
the following days. This evaluation will inform limitations in the structure of the GR4J model and
identify processes, such as short-term or long-term memory, that may require improvement.
Additionally, if potential evapotranspiration has a significant impact, it suggests that either the related
formulation in the GR4J model or the observations may require further investigation.



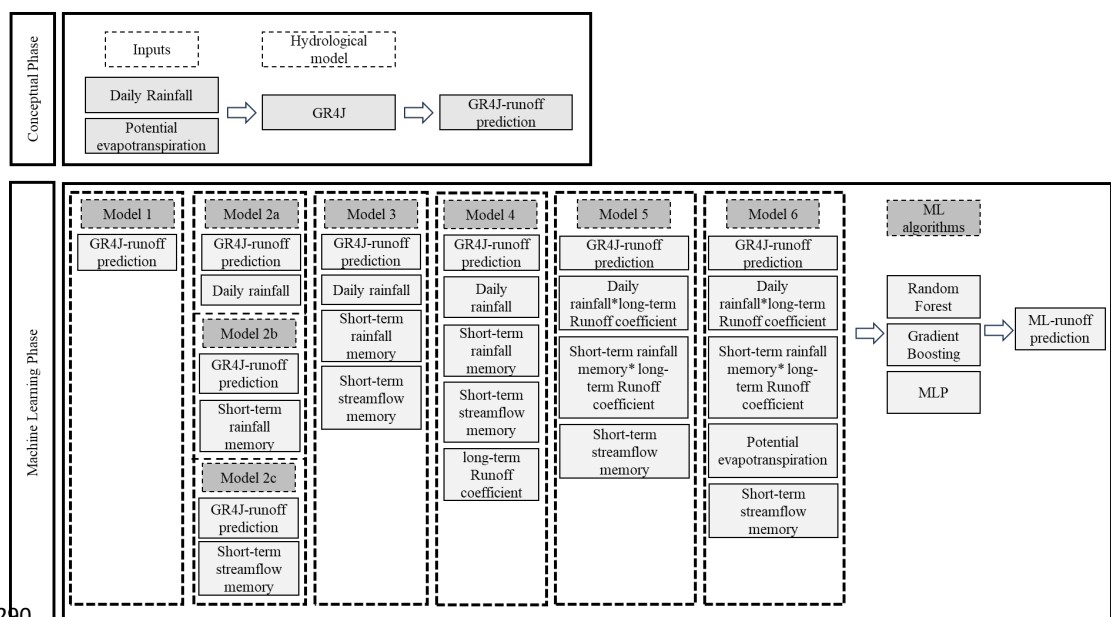

*Figure 3 Schematic of the applied methodology.*

The performance of the GR4J and ML models was assessed using the Nash-Sutcliffe Efficiency (NSE) and NSE of log-transformed flows. To assess the model performance for low-flows, the NSE of log-transformed flow was investigated, because logarithmic transformation increases the sensitivity of model performance assessment to low flow values (Romanowicz, 2007). To provide insight into the performance across the range of catchments investigated, exceedance curves for performance measures were generated. Where an ensemble of ML models is established, an ensemble of predictions is generated, one for each model, the NSE computed separately for each ensemble member and then averaged to produces a mean NSE value across the prediction ensemble.

The classification system for NSE of daily flow simulations devised by (Moriasi et al., 2015) was used to categorise model performance Table 1.

*Table 1. Model performance evaluation criteria for daily flow, based on the NSE value.*

| Very good | Good | Satisfactory | Not Satisfactory |
|---|---|---|---|
| NSE > 0.80 | 0.70 < NSE ≤ 0.80 | 0.50 < NSE ≤ 0.70 | NSE ≤ 0.50 |





Sensitivity analyses were conducted to evaluate the impact of varying two parameters: the number of
days considered for short-term pre-streamflow in model 2c and the duration of years used to calculate
the long-term runoff coefficient in model 5. Specifically, the analysis examined short-term periods
ranging from 2 to 7 days while keeping the long-term period fixed at 5 years, and separately, long-term
periods spanning 1 to 5 years with a fixed short-term period of 2 days, to assess how these variables
affect model performance and prediction accuracy.

# 3    Results and discussion

The focus of this study is to understand the additional information required to improve streamflow
simulations in a period with different rainfall-runoff characteristics for fitting models. Therefore, we
firstly assess the differences in mean daily rainfall and runoff between the model training and prediction
periods (Figure 4). The decrease in the streamflow across all sub-catchments in this region is
considerably larger than the reduction in the precipitation. This observation highlights the significance
of this region as a proper case study, prompting further investigation to understand the underlying
reasons for this discrepancy.

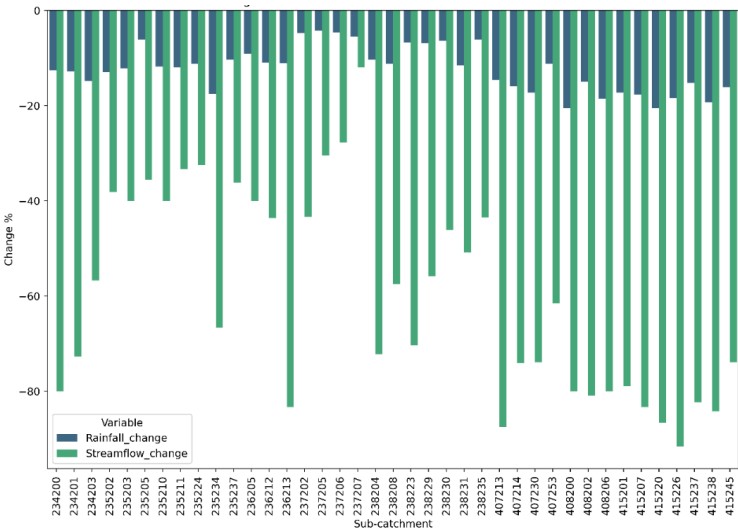

*Figure 4. Comparison of the mean daily rainfall and streamflow change from 1988-01-01 to 1996-12-31 (pre-drought) and*

*from 1997-01-01 to 2018-12-31 (drought and post-drought).*





The NSE values of the GR4J hydrological model predictions during the drought and post-drought
prediction periods are shown in Figure 5. The NSE for the prediction period are considerably lower
than the calibration period for all catchments. For the prediction period, the performance of the GR4J
hydrological model simulations can be classified as "satisfactory" for only 5 sub-catchments, with the
remainder "not satisfactory".

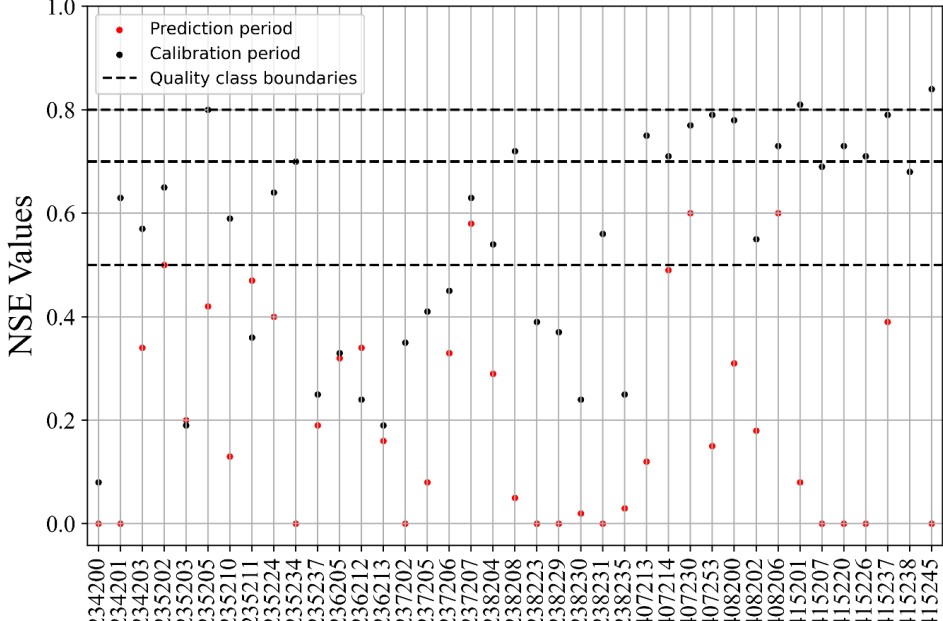


*Figure 5 NSE values of the GR4J hydrological model, for the calibration period and prediction of drought and post-drought*
*periods when calibrated to the pre-drought period. The horizontal dashed lines represent the boundary between the*
*performance classes as defined in Table 1. Negative values are shown as zero in this figure.*
The performance of the three ML algorithms across the study catchments is presented as the exceedance
probability curves in Figure 6. For each value of the NSE in the X-axis, this curve shows the percentage
of the sub-catchments that have an NSE value higher than this specific value. A larger area under the
curve thus indicates a better model performance and for enhanced clarity, negative values are omitted.
None of the models 1, 2a, and 2b show any improvement over the GR4J predictions, whereas for model
2c all ML algorithms lead to a clear improvement. Model 2c is analogous to a deterministic error
correction model that uses recent streamflow observations to update GR4J predictions. MLP, Random



Forest, and Gradient Boosting exhibit consistent improvement for both Model 5 and Model 6 in around
90% of the sub-catchments. MLP exhibits the strong overall performance for all models, which can be
attributed to the non-linear nature of the problem, where MLP excels in addressing nonlinearity. To
show and compare the performance of models with the best performing ML algorithms in each gauge,
box plots of 10 ensemble members are provided in the Appendix, Figure A. 2.
For a better understanding of the behaviour of the eight proposed models, exceedance probability curves
depicting mean NSE values are provided in Figure 7. Model 2c demonstrates improvement with respect
to GR4J, which emphasises the importance of taking into account short-term streamflow memory.
Model 3 shows improvement with respect to model 2c, which indicates the importance of short-term
rainfall and streamflow memory and daily rainfall combination. Model 5 and Model 6 clearly
outperform Model 3. While model 4 does not show improvement to model 3. This suggests that
affecting long-term runoff coefficient to rainfall and short-term rainfall memory can play a crucial role
in improving daily streamflow predictions, and therefore the GR4J model would appear to be lacking
the ability to represent long-term rainfall-runoff dynamics. Using the MLP algorithm, the performance
of Model 5 and Model 6 is relatively similar, indicating that using potential evapotranspiration as a
predictor adds little value to the predictions.





*Figure 6 Exceedance probability curves of the mean NSE values for different ML algorithms for All models.*


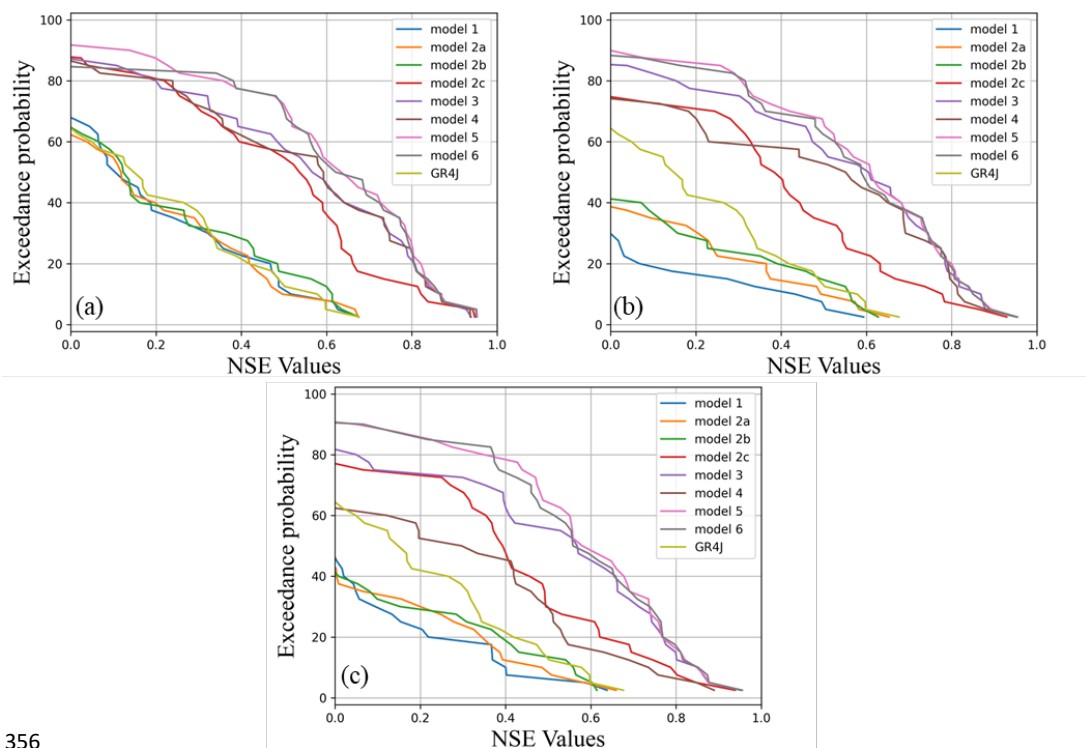

*Figure 7 Exceedance probability curves of mean NSE values of different models by application of different algorithms a)*

*MLP, b) Random forest, and c) Gradient boosting.*

In some sub-catchments, despite the considerable improvement of the NSE in comparison to GR4J, there are still unsatisfactory results, NSE values less than 0.5 for approximately 30% of sub-catchments, indicating the potential for further enhancement. This suggests that other parameters, such as NDVI, LAI, groundwater exchange, etc., should be considered as predictor in future studies.

The low-flow performance of the ML models assessed in Figure 8 and Figure 9. The exceedance probability curves for the NSE of log-transformed streamflow for the ML models and GR4J results are presented in Figure 8. All ML algorithms produce more accurate low flow predictions than GR4J. In contrast to the results for NSE, Random Forest demonstrates the greatest ability to improve low-flow predictions. Model 2c showed generally better performance compared to the other models when using the MLP algorithm. Models 3, 5, and 6 performed similarly with the Random Forest algorithms and better than models 2c and 4. When Gradient Boosting was applied, both Model 5 and Model 6 yielded



similar results, surpassing other models, while model 4 showed very poor performance. These
enhancements underscore the effectiveness of machine learning and highlight how predictors and
algorithms influence low-flow streamflow predictions. Consequently, it appears that the GR4J model
lacks the capability to adequately represent long-term rainfall-runoff and short-term rainfall and runoff
dynamics, particularly in the context of low flows.

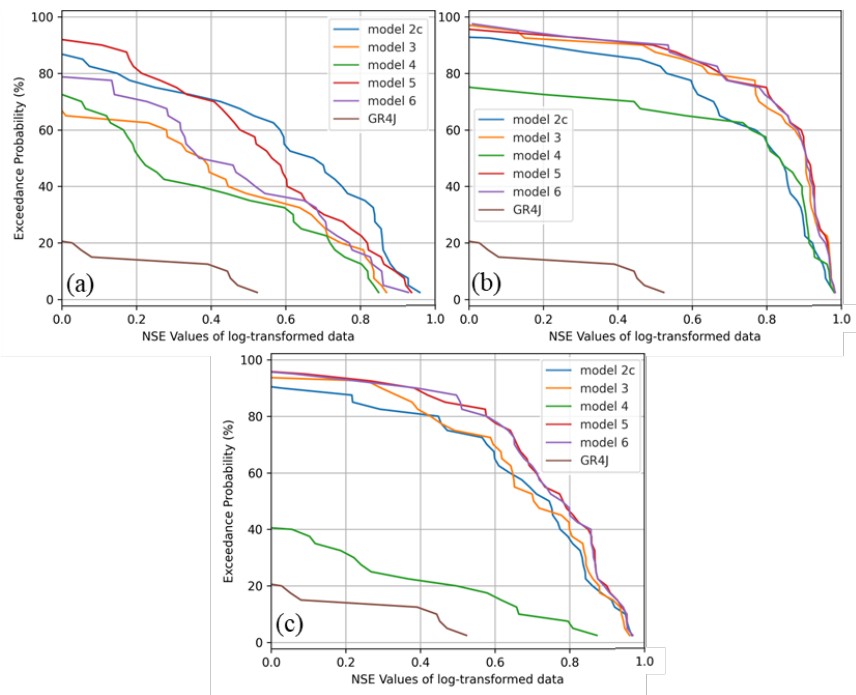


*Figure 8 Exceedance probability curves of mean NSE values of log-transformed streamflow, for different models by*
*application of different algorithms a) MLP, b) Random Forest, and c) Gradient boosting.*
Figure 9 represents the heatmap of log-transformed observations vs log-transformed predictions for
model 3, 5, and 6, as the best performed models. The colour of each cell represents the number of
observation-prediction pairs in that cell. Overall, the GR4J model tends to overestimate predictions for
lower streamflow. The application of ML algorithms resulted in more symmetrical shape compared to
GR4J, with ML predictions more closely following the identity line, indicating improved prediction
accuracy. However, in the case of MLP, the darker coloration of cells for log-transformed observations
and ML predictions below -4 suggests two issues: the former indicates overestimation of some zero-



385 flow observations, while the latter indicates no-flow predictions for days with actual streamflow. This

386 undesired behaviour was mitigated using other algorithms, with Random Forest demonstrating the best

387 performance among the candidates. Furthermore, the darker and narrower area around the equity line

388 in the Random Forest figures, particularly at lower log-transformed values, highlights significant

389 prediction improvement. However, the difference between models remains indistinct, indicating that

390 while low-flow prediction has improved, it is not evident which predictor has the greatest overall

391 impact. These results underscore the significant influence of applying ML specifically to low-flow

392 conditions, which is the primary focus of this study, i.e., the Millennium drought.









395                                Figure 9 Heatmap of the log-transformation result



To better understand the role of long-term runoff coefficient in improving streamflow predictions it is
examined how errors in GR4J simulations vary with the long-term runoff coefficient (Figure 10), and
how the MLP Model 5, as the best performed model, uses the long-term runoff coefficient to improve
streamflow predictions (Figure 11). Overall, the GR4J streamflow simulations cover a wide range of
values for all ranges of the long-term runoff coefficient, although few low streamflow values simulated
when the long-term runoff coefficient is high. Errors tend to be small for low GR4J streamflow
simulations, however for higher streamflow simulations the errors vary with the value of the long-term
runoff coefficient. For values of the long-term runoff coefficient (0 to 0.4), GR4J underestimates lower
flows and overestimates higher flows. In the 0.4–0.5 range, GR4J predominantly shows
overestimations. However, there are no clear patterns for long-term runoff coefficients in the 0.5–0.7
range, where GR4J simulations exhibit small errors. These results indicate that the magnitude of errors
in simulated streamflow vary with the long-term runoff coefficient, particularly for simulations of
higher streamflow.
The ML method corrects the streamflow simulations to reduce these error patterns, by increasing
simulations of large streamflow where the long-term runoff coefficient range (0.0 to 0.4). The ML
method makes little correction to streamflow predictions when the long-term runoff coefficient is larger
than 0.5. The corrections related to the long-term runoff coefficient being applied by the ML method,
to the GR4J simulations, are in addition to corrections related to short-term streamflow memory.
Therefore, the results are suggesting that GR4J simulation errors are dependent on the prevailing long-
term hydrological conditions, and specifically related to the ability of GR4J to simulate high streamflow
during dry conditions. Therefore, the analysis provides insights into structural weaknesses in the GR4J
model for the catchments investigated in this study.



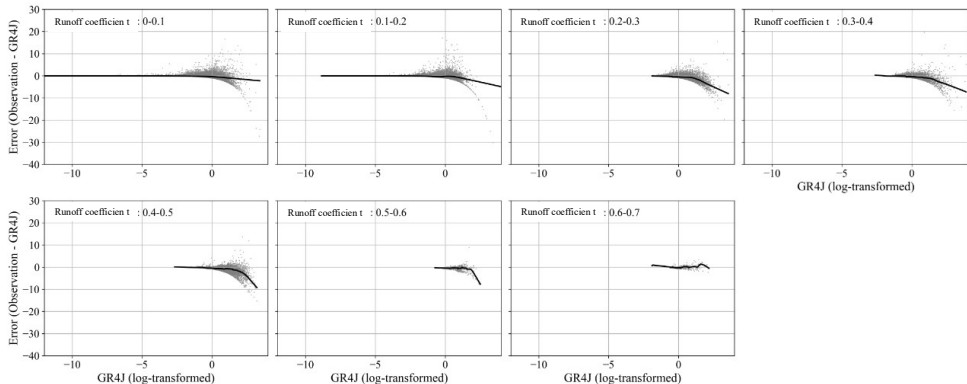


*Figure 10 GR4J error categorized based on long-term runoff coefficient ranges. Solid line represents a LOWESS (locally weighted scatterplot smoothing) line of best fit.*


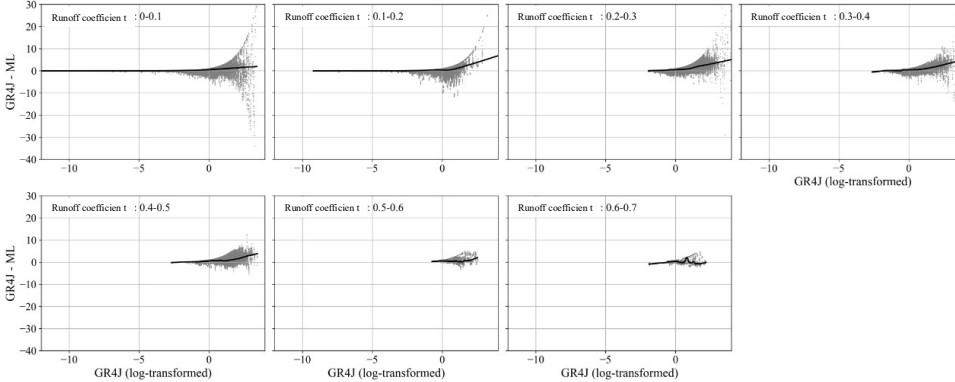


*Figure 11 GR4J and ML(MLP model5) difference categorized based on long-term runoff coefficient ranges. Solid line represents a LOWESS (locally weighted scatterplot smoothing) line of best fit.*


Sensitivity analysis of the short-term streamflow memory predictor to the duration over which the
predictor is computed indicates that shorter durations lead to slight improvements in NSE values (Figure
A. 3). This improvement is likely due to the increased relevance of recent streamflow conditions in
error correction and model prediction. In contrast, for the long-term runoff coefficient, the results show
no significant differences, and no clear pattern emerges between different durations (Figure A. 4). This
lack of consistency suggests that the influence of the long-term runoff coefficient is highly dependent
on the specific characteristics of the sub-catchment, with variations observed across different sub-
catchments.






## 4 Conclusion

This study has investigated streamflow prediction before, during and after the Millennium drought
(1997 to 2009) in catchments of Western Victoria, Australia. The GR4J hydrological model was found
to produce poor streamflow predictions for the drought and post-drought periods when calibrated to
pre-drought conditions. Multiple machine learning methods were used to investigate possible
improvements to streamflow predictions, using GR4J model simulations and physical data as predictors.
Eight combinations of predictors were considered. All ML algorithms investigated improved the
streamflow prediction for the drought and post-drought period in comparison to GR4J when trained
using data for the pre-drought period. MLP, Gradient Boosting, and Random Forest produced
simulations with greater accuracy overall and for low flow conditions than Linear Regression, Ridge
Regression, SVR, and Decision Tree. The better performing ML methods have greater ability to
characterise non-linear relationships between predictors and streamflow, than the poorer performing
methods, suggesting that the calibrated GR4J model inadequately resolving some non-linear rainfall-
runoff processes in the study sub-catchments.
Model 5 that combines predictors representing short-term memories and product of long-term runoff
coefficient and rainfall without considering potential evapotranspiration as predictor showed the
greatest improvements in prediction accuracy. This model improved streamflow predictions for around
90% of the sub-catchments using MLP, Random Forest, and Gradient Boosting algorithms. The
improvements in streamflow predictions arising from the use of long-term runoff coefficient as
predictors for the machine learning models suggests that the calibrated GR4J models inadequately
represent the slow or long-term dynamics of rainfall-runoff processes. These slow dynamics would
therefore appear to be critical for modelling low streamflow, particularly those that occur during drought
periods.
In this study, models were training using only data for the pre-drought period that was wetter than the
drought and post-drought periods on which their performance was assessed. The calibrated GR4J



models showed considerably poorer performance during the drought and post-drought periods.
However, the predictions from the ML models, when trained using data for the same period, were able
to produce predictions for the drought and post-drought periods that are considerably more accurate
than the GR4J model. The better performing ML models used a combination of the GR4J simulations
and long-term/short-term variables as predictors. The improved performance of the ML model
predictions for the post-drought period relative to those of GR4J, indicate that the long-term and short-
term streamflow memory effects exist in the pre-drought period. Therefore, it should be possible to
introduce modifications to the GR4J model, used here, to better represent the dynamics related to long-
term and short-term variables. It is found that consideration of long-term runoff coefficient and
application of ML are most suitable for catchments with low runoff coefficient. As such, this study
underscores the important role that ML methods can play in identifying directions for improving models
used for streamflow prediction, particularly under variable climatic conditions in drought-prone
regions.
**Authorship contribution statement**
Arash Aghakhani: Conceptualization, Data curation, Formal analysis, Methodology, Investigation,
Software, Validation, Writing - original draft, Writing - review & editing
David Robertson: Conceptualization, Data curation, Formal analysis, Methodology, Funding
acquisition, Resources, Supervision, Validation, Writing - review & editing
Valentijn Pauwels: Conceptualization, Formal analysis, Methodology, Funding acquisition,
Supervision, Validation, Writing - review & editing
**Declaration of Competing Interest**
"The authors declare that they have no conflict of interest."
**Data availability**
Code will be available on GitHub. HRS stations data are available at (Hydrologic Reference Stations,

484  2024)



**Acknowledgement**
This work was conducted on the traditional lands of the Boonwurrung peoples of the Kulin Nation. We
acknowledge their continuing custodianship of these lands and the rivers that flow through them, and
pay our respects to their elders, past and present. We also acknowledge the traditional custodians of the
catchments and rivers used in this study. This research has been supported by Monash University and
CSIRO.



# 5  Appendix

*Figure A. 1 Geographical representation of study area*




*Table A. 1 Sub-catchments general information*

| Sub-catchment | Station No. | Gauge Name | Area of sub-catchment (km2) | Catchment Name |
|---|---|---|---|---|
| 1 | 236213 | Mena Park | 448.3 | Hopkins River |
| 2 | 407214 | Clunes | 299.9 | Loddon River |
| 3 | 407230 | Strathlea | 156 | Loddon River |
| 4 | 408200 | Coonooer | 2677.3 | Avoca River |
| 5 | 415226 | Carrs Plains | 124.9 | Avon River-Tyrell lake |
| 6 | 234201 | Cressy (Yarima) | 1166.5 | Barwon River-Lake Corangamite |
| 7 | 234203 | Pirron Yallock (above H'Wy | 161.3 | Barwon River-Lake Corangamite |
| 8 | 235203 | Curdie | 721.3 | Otway Coast |
| 9 | 235224 | Burrupa | 1042 | Otway Coast |
| 10 | 235234 | Gellibrand | 76.6 | Otway Coast |
| 11 | 235237 | Curdie (Digneys Br) | 340.3 | Otway Coast |
| 12 | 236205 | Woodford | 893.1 | Hopkins River |
| 13 | 236212 | Cudgee | 225.4 | Hopkins River |
| 14 | 237202 | Heywood | 268.1 | Portland Coast |
| 15 | 237205 | Homerton Br | 713.7 | Portland Coast |
| 16 | 237206 | Codrington | 474 | Portland Coast |
| 17 | 237207 | Heathmere | 312.4 | Portland Coast |
| 18 | 238204 | Dunkeld | 384.9 | Glenelg River |
| 19 | 238223 | Wando Vale | 180.2 | Glenelg River |
| 20 | 238229 | Chetwynd | 69 | Glenelg River |
| 21 | 238230 | Teakettle | 197 | Glenelg River |
| 22 | 238231 | Big Cord | 54 | Glenelg River |
| 23 | 238235 | Lower Crawford | 600.8 | Glenelg River |
| 24 | 407213 | Carisbrook | 471.5 | Loddon River |
| 25 | 408206 | Archdale Junction | 760.6 | Avoca River |
| 26 | 415201 | Glenorchy Weir Tail Gauge | 1976.1 | Wimmera River |
| 27 | 415220 | Wimmera HWY | 543 | Avon River-Tyrell lake |
| 28 | 415238 | Navarre | 139.7 | Wimmera River |
| 29 | 415245 | Crowlands | 159.2 | Wimmera River |
| 30 | 407253 | Minto | 652.1 | Loddon River |
| 31 | 415237 | Stawell | 244.2 | Wimmera River |
| 32 | 415207 | Eversley | 304.5 | Wimmera River |



| Sub-catchment | Station No. | Gauge Name | Area of sub-catchment (km2) | Catchment Name |
|---|---|---|---|---|
| 33 | 408202 | Amphitheatre | 82.6 | Avoca River |
| 34 | 234200 | Pitfield | 316.9 | Barwon River-Lake Corangamite |
| 35 | 235202 | Upper Gelliberand | 52.7 | Otway Coast |
| 36 | 235210 | Gellibrand | 53.9 | Otway Coast |
| 37 | 235205 | Wyelangta | 4.5 | Otway Coast |
| 38 | 235211 | Kennedys Ck | 263.7 | Otway Coast |
| 39 | 238208 | Jimmy Creek | 23.3 | Glenelg River |


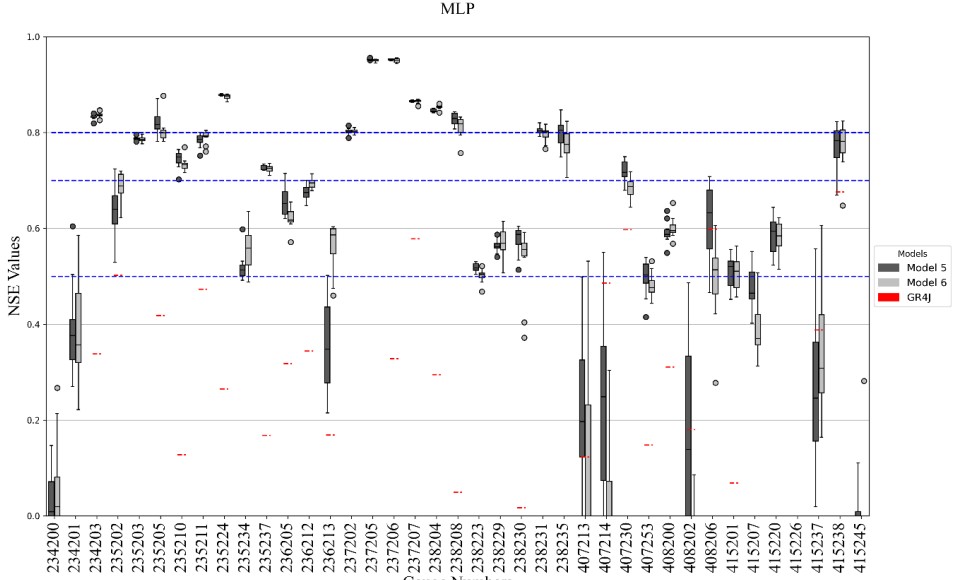




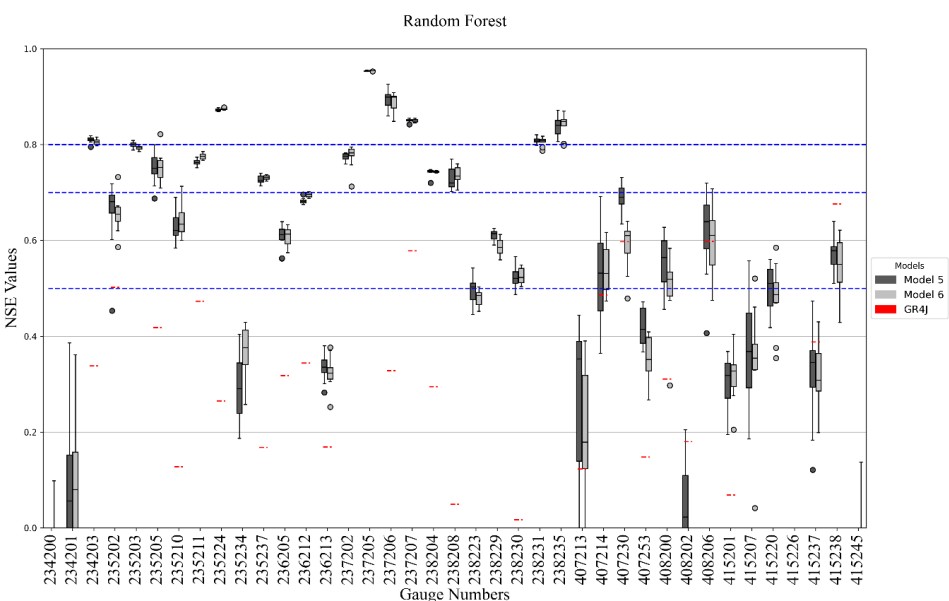



*Figure A. 2 box plots of NSE values of 10 repeats*






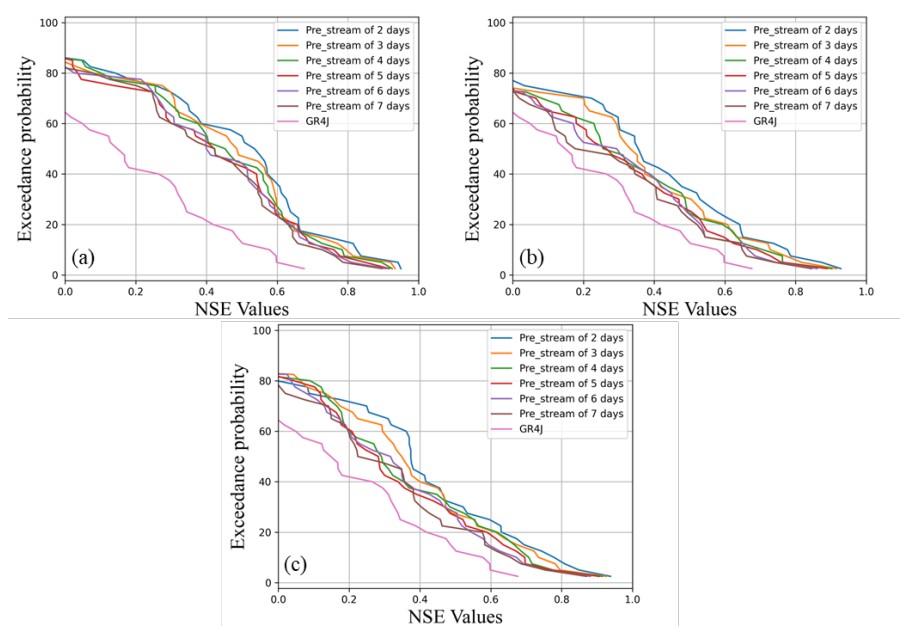

Figure A. 3 Sensitivity analysis of short-term pre-stream (days), a) MLP, b) Random Forest, and c) Gradient boosting.

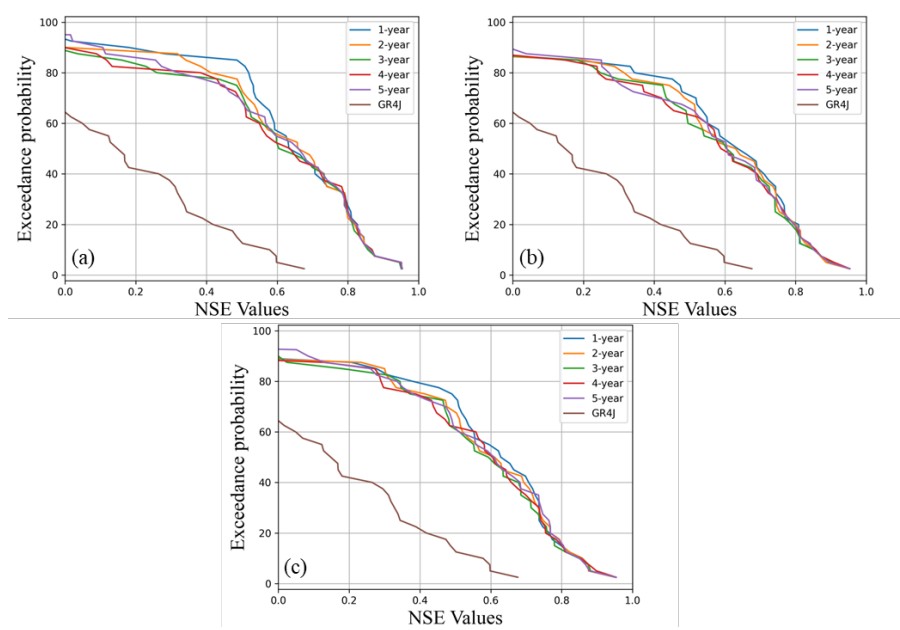

Figure A. 4 Sensitivity analysis of Long-term runoff coefficient (years), a) MLP, b) Random Forest, and c) Gradient boosting



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
