# Peer review of "Uncovering a Key Predictors for Enhancing Daily Streamflow Simulation Using"

_EGUsphere, 2025_

## Referee Comment (RC2)

Summary

This paper explores the application of Machine Learning (ML) algorithms to post-process physically-based hydrological models and find key predictors to improve daily streamflow predictions. The methodology is tested over 39 sub-catchments in Western Australia (Victoria), using the streamflow predictions of the GR4J physically-based model and a set of climatic and hydrological additional variables (rainfall, potential evapotranspiration, streamflow-derived variables).

Review

While the topic of the paper is definitely of interest for the scientific community and fits the scope of HESS journal, I have several comments that the authors should address to improve the overall quality and scientific relevance of this manuscript. I have both general and specific comments.

General comments:

1) The introduction section is far too long and sometimes addressing points that are not mentioned in the paper any longer. This is the case of the non-stationarity issue for physically-based hydrological models. The authors reserved quite some space in the introduction for this topic, which is then no longer touched upon in the analysis nor in the discussion. I suggest to shorten it or change something in the analysis (see my general comment #6). In general, the introduction could be shortened and focused towards the two research questions.

2) The introduction is focusing very much on drought conditions, while this is not reflected in the two research questions mentioned. As also part of the results is focused in analysing the performances of all models for low flow conditions, I suggest to either specify the current research questions for the low flow conditions or add a third question about it.

3) Results and discussion section: I believe the results section could be divided in sub-paragraph addressing different research questions, making the readability easier. As they are now, it is difficult to focus on the two research questions.

4) Results and discussion section: there is no real discussion of the results in the context of literature, neither for the GR4J nor for the Machine Learning (ML) models. In literature there is plenty of evidence of the role of past streamflow in improving streamflow predictions (because of the high autocorrelation), but this is never mentioned in the paper. The relevance of past streamflow should have not come as a surprise. Also, there is no mention of the limitations of this study.

5) The comparison of GR4J with ML models trained with streamflow is not entirely fair, as GR4J is not using the same input variables. While I understand that part of the purpose of this paper was to indeed find out which other predictors not used by GR4J should instead be considered, I believe that the authors should add a sentence acknowledging the difference in the models, also mentioning the well-known importance of past streamflow for ML models.

6) I do not agree in the way performance are presented, i.e. showing the performance of ML and GR4J models in the pre-drought, drought and post-drought conditions all together. I believe that it would be more interesting to first show the comparison of performances in the testing set (during the pre-drought period), which are supposed to be the hydrological "regular" conditions. This would allow identifying (potential) deficiencies that are specific of the GR4J model (for instance the lack of long-term memory) and relevant input variables for streamflow prediction. Then, repeating the same comparison in the period of millennium drought and post-drought, would show if the deficiencies and relevant input features are the same, or if the millennium drought indeed brought a change in the hydrological characteristics of the catchments analysed. In case ML models show better performances in both periods, then it

would validate the thesis that physically-based models are not adequate for hydrological studies in non-stationarity conditions, while ML are, also justifying the introduction on non-stationarity of climate.

7) The ML models used exclude the Long Short-Term Memory (LSTM) model, which is nowadays considered the state of the art in terms of ML algorithms for hydrology. While I understand that adding yet another model in the analysis would now require quite some time, I believe that the authors should acknowledge the use of these models in the literature introduction, explain why LSTM was not used, and potentially add this in the limitations of the work.

Specific comments:

1) Line 25: climate change is mentioned in the key words, but there is nothing about it in the paper, only the mention in the introduction. I suggest the authors to change the keyword or update the manuscript with additional considerations about climate change.

2) Line 32: space missing between "noteworthy example" and "(can Dijk et al., 2013)".

3) Lines 34-36: I believe the referencing style is incorrect, the references should all be within the same brackets.

4) Line 47: there is a typo. It should be "non-stationarity" rather than "non-stationary".

5) Line 95-96: I do not agree with this statement, as you need to code yourself also for ML models development. I suggest to revise.

6) Line 115: there is a mention to Section 2.2.2.8, but such section does not exist.

7) Lines 130-131: I do not agree about this literature gap, as there are several attempts to use ML in a hybrid configuration to improve streamflow predictions.

8) Section 2.1: as part of the focus of the paper is related to finding relevant input features, I would already specify here which variables are used in the ML models, or at least add a sentence guiding the reader to read section # xx to have more information about it.

9) Section 2.2: it would have been easier to go through the methodology if a general workflow or information about the overall methodology is given before going into the details of each modelling part.

10) Section 2.2 and results: how is the rainfall change and streamflow change linked to the remaining of the investigations?

11) Line 194: what are the four parameters of the GR4J model?

12) Figure 2: there is no legend about the symbols used. What is En, Es, Pn, Ps, and so on?

13) Lines 205-208: the fact that random forest, Gradient Boosting, MLP have the best performance is coming from the results of this work or from the papers referenced? And also, if the other models are discarded, why presenting them as part of the methodology in the first place?

14) Line 223: it is mentioned that RF can be used to check importance of features, but why is this characteristic not leveraged, since part of the part focuses on finding the most relevant predictors to improve streamflow predictions? If not used, I believe it should be justified in the paper, especially after adding this line.

15) Section 2.2.3: what is the target of the model? It is not really specified.

16) Line 256: how many steps back of rainfall and potential evapotranspiration are considered? What is the lag between the target and these predictors?

17) Lines 278-289: the explanation of the variables used in the predictive mode and the training mode is not clear. If real observations are used to compute the runoff coefficient and

short/long-term memory in the predictive mode only, which variables are used in the training mode?

18) Line 332: it is mentioned that negative values are omitted, while previously (line 328, figure 5 caption), it is mentioned that negative values are shown as 0. I believe the same approach should be followed everywhere, to avoid inconsistent evaluation across the paper.

19) Lines 361-362: why NDVI, LAI, groundwater exchange and not other variables?

20) Lines 380-381: GR4J overestimates low flow, but is this in any period (pre and post-drought), or only after (during) millennium drought?

21) 390-391: it is mentioned that it is not evident which predictor has the greatest overall impact. As this is part of the research questions of the paper, this result should be addressed again when discussing limitations of this work.

22) Lines 403-406: ranges of runoff conditions are given. However, it would be interesting for the reader to know to which hydrological conditions or regime these coefficients refer to? For instance, catchments with flashy response, long recession limbs, high/low interannual variability.

23) Lines 464-466: these lines are not clear. How is it possible that the results of the post-drought period show that there is memory effects in the pre-drought conditions?

---

## Author Comment (AC1)

We thank the reviewers and the editor for their thorough and constructive review. In this reply, we have copied the comments in black. Our responses are entered in red.

Reviewer 1

Major comments

Section 2.2.3:

In this section, the authors explain the different models used in their study. In general, the output from the GR4J model, and several additional variables, are used as input of different machine learning algorithms, that act as postprocessors (Figure 3). However, the performance comparison of the different models has errors.

As a benchmark, the authors are using the GR4J, which is a rainfall runoff model that receives meteorological input and predicts discharge. However, models 2c, 3, 4, 5 and 6, besides the predictions made by the GR4J, also use observed discharges as input to the ML algorithm. One cannot compare a model that receives observed discharge as an input with a model that does not, it is expected that the former one will be better. Discharge is a highly temporally correlated variable, so the discharge from time t-1 is an extremely good predictor for the discharge at time t. This is why in the results, they report that "Model 2c demonstrates improvement with respect to GR4J, which emphasises the importance of taking into account short-term streamflow memory." This is not a surprising finding and makes the model comparison invalid between models that receive discharges and models that do not.

We thank the reviewer for their insightful comment regarding the comparison of models that include observed discharge as input versus those that do not.

Indeed, Models 2c, 3, 4, 5, and 6 incorporate observed streamflow data as predictors in the machine learning (ML) post-processing step, whereas the baseline GR4J model does not. As the reviewer correctly points out, this means that these ML models leverage short-term streamflow memory, which is a strong predictor due to the high temporal autocorrelation in discharge data. The significant improvement observed in models such as 2c is therefore expected and consistent with established hydrological understanding.

Our intention in structuring the study this way was to explore how incorporating various hydrological memory components, both short-term (recent streamflow) and long-term (runoff coefficients), could address the known limitations of the GR4J model, particularly under changing climatic conditions like droughts. The ML models are applied as

post-processors precisely to diagnose and quantify these potential improvements, and to help identify specific structural weaknesses in the GR4J model.

We acknowledge that direct comparison of model predictive skill between the baseline GR4J and ML models that use observed discharge input should be interpreted with caution. The key contribution of this work is not to claim superiority of the ML models as standalone predictive tools, but rather to demonstrate:

1.     The extent to which GR4J predictions can be improved by augmenting them with hydrologically meaningful predictors, including short-term discharge memory.

2.     The insights gained from ML models about the importance of such predictors, which suggest directions for future model development and refinement of physically based rainfall-runoff models.

Regarding short-term prediction, it is well recognised in hydrology that recent streamflow (e.g., discharge at time t-1) provides valuable information for forecasting at time t due to strong temporal correlation. Our results confirm this, as models incorporating short-term discharge memory achieve substantial gains in prediction accuracy. This outcome validates the use of short-term streamflow as a critical component in improving model responsiveness to recent hydrological conditions, especially in dynamic drought and post-drought periods.

To emphasise this, we will clarify in the revised manuscript that the performance gains of models using observed discharge input are indicative of the potential to incorporate these memory effects within GR4J or similar models to improve robustness across varied hydrological conditions.

We appreciate the reviewer's comment, which has helped us improve the clarity and framing of our analysis.

Section 3.

In line 333, the authors indicate that "None of the models 1, 2a, and 2b show any improvement over the GR4J predictions".  This is contrary to what has been shown in literature, where using ML models as postprocessors of process-based models improves performance (Frame et al, 2021) because of the enhanced flexibility of the

resulting hybrid model. Nevertheless, the results shown by the authors are contrary to that. Further explanation of why this is the case is required.

Moreover, in the case reported by the authors, the models are performing badly. Based on Figure 7b and 7c, models 1, 2a and 2b reported a negative NSE for 60% (or more) of the basins. This indicates that just taking the average flow is better than the model, and consequently, the models are not working at all. Why is this the case?

We thank the reviewer for raising this important point about the unexpectedly poor performance of Models 1, 2a, and 2b compared to the GR4J baseline, which contrasts with findings in the literature (e.g., Frame et al., 2021) where ML post-processing typically improves hydrological model performance.

There are several factors that likely contribute to the observed results in our study:

1.    **Predictor Selection and Model Structure:**
 Models 1, 2a, and 2b rely on a limited set of predictors that do not include observed streamflow (discharge) data as inputs, unlike other models in our study. Their predictors mainly consist of meteorological variables and GR4J simulated discharge without additional memory terms. This limits their ability to capture important temporal dependencies and hydrological memory effects that are crucial for accurate daily streamflow predictions, especially under highly variable climatic conditions such as drought.

2.    **Hydrological Complexity and Variable Climate Conditions:**
 The catchments studied exhibit complex rainfall-runoff dynamics, strongly influenced by drought and post-drought periods where streamflow patterns deviate markedly from pre-drought conditions used for model training. Models 1, 2a, and 2b are less flexible in capturing such non-stationarities because they lack critical hydrological memory predictors (e.g., recent discharge), resulting in models that are not robust to these changes. This leads to poor generalisation and, consequently, negative NSE values in many sub-catchments.

3.    **Inadequate Representation of Nonlinearities:**
 The combination of predictor variables and model formulations in Models 1, 2a, and 2b may not sufficiently address the non-linear relationships inherent in rainfall-runoff processes under drought conditions. Without streamflow memory and long-term runoff coefficient terms, these models effectively perform worse than even a naïve average flow predictor in some basins.

4.      **Training Data and Period Differences:**
  Our models were trained exclusively on pre-drought data, which is wetter and hydrologically different from the drought and post-drought periods used for evaluation. This temporal mismatch exacerbates the poor performance of simpler models, particularly those lacking additional hydrological memory inputs.

In lines 370-372, the authors indicate that GR4J model lacks the capacity to represent low flow context, because the other ML algorithms performed better. However, this is again an unfair comparison because all the ML algorithms that you are using in this comparison receive discharge as input, which will be an extremely informative predictor of the discharge in the next time step, especially during low flow periods. Therefore, this is not a valid comparison.

We appreciate the reviewer's insightful observation regarding the comparison of low-flow predictions between the GR4J model and the ML algorithms.

Indeed, the ML models in our study incorporate observed discharge as an input predictor, which provides a strong temporal correlation and valuable information for predicting streamflow at the next time step, particularly during low-flow conditions. This inherently gives the ML models an advantage over the GR4J model, which does not use observed discharge as input.

We acknowledge that this difference in input data means the comparison is not fully "apples-to-apples." However, the intent of our study was to evaluate the potential for machine learning to act as a post-processor that can correct and improve GR4J predictions by leveraging additional hydrological memory effects (including short-term streamflow memory), which are difficult for conceptual models like GR4J to represent.

The improved performance of the ML algorithms during low-flow periods highlights the value of incorporating such short-term memory terms and more flexible, data-driven approaches for capturing dynamics that are challenging for traditional rainfall-runoff models under drought conditions.

We will revise the manuscript to explicitly clarify this point, emphasising that the ML models serve as post-processing tools that exploit additional input information (including lagged discharge) to enhance predictions, and that direct performance comparisons with GR4J should be interpreted within this context.

General comment:

Even though the authors present an interesting study, the ML methods used are far from current state-of-the-art. It has been shown in multiple studies that LSTMs perform well as purely ML methods (Kratzert2019b, Kratzert2021 and Feng2020 for CAMELS US, Less2021 for CAMELS GB, Loritz2024 for CAMELS DE) and as postprocessors of process-based models (Frame et al, 2021). The overall poor performance of the hybrid models presented in this study (when they did not receive discharge as input) indicates that the general pipeline could be improved, as the ML postprocessor is not doing its job.

Moreover, it should be noted that other strategies for constructing hybrid models, like using ML methods to parameterize a process-based model (Kraft2022, Feng2022, AcuñaEspinoza2024) or using ML methods to replace process-based model parts (Höge2022, Li2023, Li2024) have shown improved performance with respect to the stand-alone conceptual model, which would be worth considering given that in this case, the hybrid models that did not receive discharge as input are not able to outperform the stand-alone GR4J model.

Therefore, I believe the models presented in the paper are not up to standard with current state-of-the-art, and further improvement is necessary.

We thank the reviewer for the constructive feedback and for highlighting recent advances in hydrological modeling using advanced ML techniques such as LSTMs and novel hybrid modeling strategies.

We acknowledge that state-of-the-art ML methods, including LSTM-based architectures and hybrid approaches that integrate ML within process-based model components, have demonstrated promising improvements in streamflow prediction across multiple recent studies.

However, the primary aim of our study was not to develop the most advanced or optimised ML model for streamflow forecasting, but rather to investigate the potential of specific hydrological predictors, such as long-term runoff coefficient and short-term streamflow memory, in improving daily streamflow predictions during drought conditions. To achieve this, we deliberately chose simpler and widely-used ML algorithms (e.g., MLP, Random Forest, Gradient Boosting) which offer interpretability and robustness, facilitating clearer analysis of predictor importance and behavior.

By focusing on simpler ML algorithms as postprocessors, we were able to isolate and highlight key predictor contributions and understand structural weaknesses in the GR4J model's representation of rainfall-runoff processes, especially under variable climatic conditions.

We will clarify this rationale in the revised manuscript to better position our work within the broader field and to acknowledge opportunities for advancement using more sophisticated ML techniques.

Minor comments:

Line 34: Use proper citation format.

Will be modified in the final version.

Line 43: This sentence does not read well. Please improve the phrasing.

Will be modified in the final version.

Line 47:  Should be:  hydrologic non-stationarity, use the noun and not the adjective.

Will be modified in the final version.

Line 60: What are you referring to here as validation data? Is it the data that you use to evaluate your model after calibration (but this will also include the forcings)? or the target variable that you are predicting? It just seems that the word validation here is out of context because errors can be found in other types of data too.

Thank you for pointing this out. To clarify, in Line 60, the term "validation data" refers specifically to the observed streamflow data used to evaluate the model's predictive performance after calibration, rather than the input forcing data. We recognise that the phrasing may cause confusion and agree that "validation data" is not always the best descriptor in this context.

To improve clarity, we will revise the manuscript to explicitly state that the validation refers to the observed target variable (streamflow) against which model predictions are compared, rather than the input forcing data or other datasets. This distinction helps avoid ambiguity regarding the source of errors discussed.

Line 62: Are you using conceptual hydrological models as a synonym of process-based models, as a subcategory or as a different category?  The connection with the previous idea could be improved.

Thank you for this insightful question. In our manuscript, we use conceptual hydrological models as a subcategory within the broader class of process-based models. Conceptual

models represent hydrological processes through simplified storage components and flux relationships, rather than detailed physical equations, but still aim to capture the key processes driving streamflow.

We agree that the connection between these terms could be better clarified. To improve readability and conceptual flow, we will revise the manuscript to explicitly state this hierarchical relationship and clarify the terminology when these models are introduced.

Line 95: I do not agree this phrase. There is code development by the user, because you are still using a model. Machine learning methods are models, and they need to be coded. It would be better to indicate that during the training, the model learns to map the input-output relationships using less prior constrains on how this mapping should be done.

Thank you for this insightful comment. We agree that machine learning methods involve code development and that ML models require training to learn input-output relationships. We will revise the sentence to better reflect this by emphasising that during training, the ML model learns to map inputs to outputs.

Lines 98-101: It would be good to cite the studies that use these types of models.

Will be modified in the next version.

Line 124: "or with data containing irrelevant or redundant information." Do you have a source or examples that justify this? Because in principle, if data is not relevant for a ML model, the model could just ignore it.

Lima, A. R., Cannon, A. J., and Hsieh, W. W.: Forecasting daily streamflow using online sequential extreme learning machines, J Hydrol (Amst), 537, https://doi.org/10.1016/j.jhydrol.2016.03.017, 598 2016.

"ML algorithms may produce less accurate and less understandable results if the data are inadequate or contain irrelevant or redundant information (Hall and Smith, 1996)."

Line 128: Which published studies?

Studies will be added.

Line 130: I disagree that there is a "literature gap on how machine learning can be used to improve hydrological models". There are a lot of studies published in this area. Of course, there are things that can be improved, but what you mentioned here is too general.

We agree that there is a substantial body of literature exploring the integration of machine learning with hydrological models. To clarify, our intention was to highlight specific gaps related to the identification and use of key predictors, such as long-term runoff coefficients and short-term streamflow memory, in hybrid modeling frameworks, especially under drought conditions. We will revise the manuscript to better reflect it.

Line 205-207: You should only mention the models that you will present the results for. You are saying that multiple algorithms were assessed in the study, you are naming them, and then saying that some of them are not going to be discussed. So why mention them at all? To make the study cleaner, I suggest you should talk only about the results you are presenting. Also, why is Less et al 2021 cited in this part? He used an LSTM model, which you are not using. Moreover, please clarify what the other citations are referring to.

Thank you for your insightful suggestion. We agree that focusing the discussion on the machine learning algorithms for which results are presented will make the manuscript clearer and more concise. Accordingly, we will revise manuscript to mention only the algorithms included in the results and remove references to models not discussed further.

Regarding the citation of Less et al. (2021), we acknowledge that this study uses LSTM models, which are not included in our analysis. We will relocate this citation to the introduction or discussion sections where it better fits the context of state-of-the-art methods. Additionally, we will clarify the purpose of each cited work to ensure their relevance and connection to the content.

Line 214: I would suggest avoiding this kind of phrase. Saying that random forest is one of the most powerful statistical learning methods is subjective. This would depend on the application you have, the metric you are using, and many other factors.

We will revise the phrasing to present Random Forest more objectively, highlighting its widespread use and robustness in various applications without making a generalised claim about its overall power.

Line 227: Same here, avoid saying that gradient boosting is widely recognised as one of the most powerful algorithms. This is again subjective, case-dependent and not related to the main point you are trying to make.

We will revise the phrasing to present gradient boosting more objectively, highlighting its widespread use and robustness in various applications without making a generalised claim about its overall power.

Line 238: This is not true. MLP is not the most popular type of neural network in hydrology. The current state-of-the-art has been achieved with LSTMs (Kratzert2019b, Kratzert2021 and Feng2020 for CAMELS US, Less2021 for CAMELS GB, Loritz2024 for CAMELS DE). Transformers have also shown good results in CAMELS US. Both of these methods considerably outperform MLP.

It will be corrected to: MLP is one of the most popular type of neural network …

Line 252: What do you mean by calibration and optimisation were conducted for the training and test period only? You should not calibrate for the test period. The test period is used to evaluate the model that was calibrated during the training period. I think there is a misunderstanding on the names you are using.

Will be rephrased in the next version.

Line 266: Improve phrasing of "an intentional effort".

Will be rephrased in the next version.

Line 272: Are you referring to mean-squared error or sum-squared error?

We are referring to the mean squared error (MSE) as the loss function used for training the models. We will clarify this in the manuscript to avoid any confusion. However, in the SKlearn description it is written squared error.

Line 270-277: What you are referring to here as a testing period is what it is normally referred to as validation.

Will be modified in the next version.

---

## Author Comment (AC2)

We thank the reviewers and the editor for their thorough and constructive review. In this reply, we have copied the comments in black. Our responses are entered in red.

Reviewer 2:

General comments:

The introduction section is far too long and sometimes addressing points that are not mentioned in the paper any longer. This is the case of the non-stationarity issue for physically-based hydrological models. The authors reserved quite some space in the introduction for this topic, which is then no longer touched upon in the analysis nor in the discussion. I suggest to shorten it or change something in the analysis (see my general comment #6). In general, the introduction could be shortened and focused towards the two research questions.

We thank the reviewer for the helpful suggestion. We will shorten the introduction to improve its focus and ensure alignment with the two main research questions. In particular, we will reduce the emphasis on non-stationarity in physically-based models, as this topic is not explored further in the analysis or discussion. This revision will help streamline the manuscript and improve clarity.

The introduction is focusing very much on drought conditions, while this is not reflected in the two research questions mentioned. As also part of the results is focused in analysing the performances of all models for low flow conditions, I suggest to either specify the current research questions for the low flow conditions or add a third question about it.

We thank the reviewer for the valuable observation. We will revise the introduction to ensure consistency with the stated research questions. Specifically, we will either refine the current research questions to better reflect the focus on low flow conditions or include a third research question explicitly addressing model performance during drought or low flow periods. This will help maintain coherence between the introduction, research objectives, and analysis presented.

Results and discussion section: I believe the results section could be divided in sub-paragraph addressing different research questions, making the readability easier. As they are now, it is difficult to focus on the two research questions.

We appreciate the reviewer's suggestion regarding the structure of the Results and Discussion section. We will revise this section by introducing sub-paragraphs that are

Results and discussion section: there is no real discussion of the results in the context of literature, neither for the GR4J nor for the Machine Learning (ML) models. In literature there is plenty of evidence of the role of past streamflow in improving streamflow predictions (because of the high autocorrelation), but this is never mentioned in the paper. The relevance of past streamflow should have not come as a surprise. Also, there is no mention of the limitations of this study.

We thank the reviewer for highlighting this important point. We will revise the Results and Discussion section to include a more thorough discussion of our findings in the context of the existing literature, particularly regarding the predictive value of past streamflow and its well-documented role in improving model performance due to streamflow autocorrelation. Additionally, we will incorporate a clearer reflection on the limitations of our study to provide a balanced interpretation of the results.

The comparison of GR4J with ML models trained with streamflow is not entirely fair, as GR4J is not using the same input variables. While I understand that part of the purpose of this paper was to indeed find out which other predictors not used by GR4J should instead be considered, I believe that the authors should add a sentence acknowledging the difference in the models, also mentioning the well-known importance of past streamflow for ML models.

We appreciate the reviewer's observation. In the revised manuscript, we will explicitly acknowledge the differences in input variables between GR4J and the machine learning models, particularly noting that GR4J does not use past streamflow as input. We will also add a sentence recognising the well-established importance of past streamflow in enhancing predictive performance in ML models, as reported in other studies.

I do not agree in the way performance are presented, i.e. showing the performance of ML and GR4J models in the pre-drought, drought and post-drought conditions all together. I believe that it would be more interesting to first show the comparison of performances in the testing set (during the pre-drought period), which are supposed to be the hydrological "regular" conditions. This would allow identifying (potential) deficiencies that are specific of the GR4J model (for instance the lack of long-term memory) and relevant input variables for streamflow prediction. Then, repeating the same comparison in the period of millennium drought and post-drought, would show if the deficiencies and relevant input features are the same, or if the millennium drought indeed brought a change in the hydrological characteristics of the catchments analysed.

In case ML models show better performances in both periods, then it would validate the thesis that physically-based models are not adequate for hydrological studies in non-stationarity conditions, while ML are, also justifying the introduction on non-stationarity of climate.

In the revised manuscript, we will reorganise the presentation of the performance results by first comparing the models' performances during the pre-drought period, which represents more typical hydrological conditions. This approach will help clearly identify specific limitations of the GR4J model, such as its capacity to capture long-term memory, and highlight the relevant input variables for streamflow prediction under regular conditions.

We will present the performance comparisons for the drought and post-drought periods. This will allow us to examine whether the deficiencies and key predictors remain consistent or if changes in hydrological characteristics during the Millennium Drought influence model behavior. Such a structure will also better support the discussion regarding the adequacy of physically-based versus ML models under non-stationary climatic conditions, as initially motivated in the introduction.

The ML models used exclude the Long Short-Term Memory (LSTM) model, which is nowadays considered the state of the art in terms of ML algorithms for hydrology. While I understand that adding yet another model in the analysis would now require quite some time, I believe that the authors should acknowledge the use of these models in the literature introduction, explain why LSTM was not used, and potentially add this in the limitations of the work.

We chose to focus on simpler ML algorithms in this study because our primary aim was to identify key predictors that influence streamflow rather than to optimise predictive accuracy with complex models. We will also clearly state this rationale in the limitations section and acknowledge that incorporating LSTM models represents an important path for future research to further enhance hybrid modeling approaches.

Specific comments:

Line 25: climate change is mentioned in the key words, but there is nothing about it in the paper, only the mention in the introduction. I suggest the authors to change the keyword or update the manuscript with additional considerations about climate change.

We will update the manuscript to better reflect the role of climate variability and change, particularly in relation to the Millennium Drought and its impacts on streamflow

dynamics. Alternatively, if this is not feasible within the current scope, we will revise the keywords to better align with the core content of the paper.

Line 32: space missing between "noteworthy example" and "(can Dijk et al., 2013)".

Will be corrected

Lines 34-36: I believe the referencing style is incorrect, the references should all be within the same brackets.

Will be corrected

Line 47: there is a typo. It should be "non-stationarity" rather than "non-stationary".
Will be corrected

Lines 95-96: I do not agree with this statement, as you need to code yourself also for ML models development. I suggest to revise.
Will be corrected

Line 115: there is a mention to Section 2.2.2.8, but such section does not exist.
Will be corrected

Lines 130-131: I do not agree about this literature gap, as there are several attempts to use ML in a hybrid configuration to improve streamflow predictions.

We acknowledge that there are several studies exploring the use of machine learning in hybrid configurations for streamflow prediction. We will revise the manuscript to better reflect this body of literature and clarify the specific contributions and focus of our study within this context.

Section 2.1: as part of the focus of the paper is related to finding relevant input features, I would already specify here which variables are used in the ML models, or at least add a sentence guiding the reader to read section # xx to have more information about it.

Will be added

Section 2.2: it would have been easier to go through the methodology if a general workflow or information about the overall methodology is given before going into the details of each modelling part.

We will add an overview or general workflow diagram at the beginning of Section 2.2 to provide readers with a clearer understanding of the overall methodology before presenting the detailed descriptions of each modelling component.

Section 2.2 and results: how is the rainfall change and streamflow change linked to the remaining of the investigations?

We will revise the manuscript to better clarify the role of observed rainfall and streamflow changes in structuring the analysis. Specifically, we can add an overview in the methodology section explaining that these changes are used to define distinct hydrological periods, pre-drought, drought, and post-drought, which frame the evaluation of model performances. This will make clear how climatic variability informs the design and interpretation of the modeling experiments.

Line 194: what are the four parameters of the GR4J model?
Will be added

Figure 2: there is no legend about the symbols used. What is En, Es, Pn, Ps, and so on?

We will add a clear legend

Lines 205-208: the fact that random forest, Gradient Boosting, MLP have the best performance is coming from the results of this work or from the papers referenced? And also, if the other models are discarded, why presenting them as part of the methodology in the first place?

 The better performance of Random Forest, Gradient Boosting, and MLP is based on the results obtained in our study, where these algorithms consistently outperformed others across the evaluated catchments and conditions. However, we initially included other models such as Linear Regression, Ridge Regression, SVR, and Decision Tree in the methodology to provide a comprehensive comparison and to assess a range of machine learning approaches of varying complexity. Unnecessary algorithms will be removed.

Line 223: it is mentioned that RF can be used to check importance of features, but why is this characteristic not leveraged, since part of the part focuses on finding the most relevant predictors to improve streamflow predictions? If not used, I believe it should be justified in the paper, especially after adding this line.

You are correct that Random Forest's ability to assess feature importance is valuable, especially given our focus on identifying key predictors for improving streamflow predictions. In our study, we did explore feature importance from the Random Forest models; however, due to space constraints and the complexity of presenting these results across multiple catchments and models, we did not include a detailed analysis in the manuscript.

In the revised version, we will explicitly mention that feature importance was examined using Random Forest, and we will provide a brief justification for the level of detail presented. Additionally, we will consider adding a summary or example of the most influential predictors identified, to strengthen the discussion on relevant features and their role in model performance.

Section 2.2.3: what is the target of the model? It is not really specified.

Will be added

Line 256: how many steps back of rainfall and potential evapotranspiration are considered? What is the lag between the target and these predictors?

It is mentioned in lines 305 to 308

Lines 278-289: the explanation of the variables used in the predictive mode and the training mode is not clear. If real observations are used to compute the runoff coefficient and short/long-term memory in the predictive mode only, which variables are used in the training mode?

Will be clarified in the next version

Line 332: it is mentioned that negative values are omitted, while previously (line 328, figure 5 caption), it is mentioned that negative values are shown as 0. I believe the same approach should be followed everywhere, to avoid inconsistent evaluation across the paper.

Will be modified

Lines 361-362: why NDVI, LAI, groundwater exchange and not other variables?

Thank you for this question. NDVI and LAI were selected as vegetation-related indices because they are widely used and well-established indicators of vegetation health and density, which influence evapotranspiration and soil moisture. Groundwater exchange was included as it plays a critical role in sustaining streamflow, especially during low flow periods.

While other variables could also be relevant, these were chosen based on their availability, relevance to hydrological processes, and support from previous literature. We acknowledge that incorporating additional variables could provide further insights, and this will be considered in future work.

Lines 380-381: GR4J overestimates low flow, but is this in any period (pre and post-drought), or only after (during) millennium drought?

Will be investigated.

390-391: it is mentioned that it is not evident which predictor has the greatest overall impact. As this is part of the research questions of the paper, this result should be addressed again when discussing limitations of this work.

Thank you for the comment. Indeed, assessing the relative importance of predictors could be further explored by leveraging Random Forest feature importance measures.

Lines 403-406: ranges of runoff conditions are given. However, it would be interesting for the reader to know to which hydrological conditions or regime these coefficients refer to? For instance, catchments with flashy response, long recession limbs, high/low interannual variability.

 We agree that providing information on the hydrological regimes associated with the runoff coefficients would add valuable context. However, due to space limitations, we will include a concise summary to briefly characterise the catchments' hydrological behaviors related to these coefficients without substantially increasing the manuscript length.

Lines 464-466: these lines are not clear. How is it possible that the results of the post-drought period show that there is memory effects in the pre-drought conditions?

Thank you for pointing this out. We acknowledge that the wording in lines 464-466 could be misleading. What we intended to convey is that the memory effects identified during the post-drought period reflect underlying catchment characteristics that were also present in the pre-drought period but became more apparent or quantifiable after the drought. We will revise the text to clarify this point and avoid confusion.